# PartialFed: Cross-Domain Personalized Federated Learning via Partial Initialization

Benyuan Sun    Hongxing Huo    Yi Yang    Bo Bai

Media Technology Lab, Huawei
{sunbenyuan,huohongxing1,yangyi16,baibo3}@huawei.com

## Abstract

The burst of applications empowered by massive data have aroused unprecedented privacy concerns in AI society. Currently, data confidentiality protection has been one core issue during deep model training. Federated Learning (FL), which enables privacy-preserving training across multiple silos, gained rising popularity for its parameter-only communication. However, previous works have shown that FL revealed a significant performance drop if the data distributions are heterogeneous among different clients, especially when the clients have cross-domain characteristic, such as traffic, aerial and in-door. To address this challenging problem, we propose a novel idea, *PartialFed*, which loads a subset of the global model's parameters rather than loading the entire model used in most previous works. We first validate our algorithm with manually decided loading strategies inspired by various expert priors, named *PartialFed-Fix*. Then we develop *PartialFed-Adaptive*, which automatically selects personalized loading strategy for each client. The superiority of our algorithm is proved by demonstrating the new state-of-the-art results on cross-domain federated classification and detection. In particular, solely by initializing a small fraction of layers locally, we improve the performance of FedAvg on Office-Home and UODB by 4.88% and 2.65%, respectively. Further studies show that the adaptive strategy performs significantly better on domains with large deviation, e.g. improves AP50 by 4.03% and 4.89% on aerial and medical image detection compared to FedAvg.

## 1 Introduction

Endless collection of texts, images and videos for training data-hungry models increases potential threats to the safety systems. Any attacks to these data centers could cause billions of information leakage. A safer way is to keep the user data purely local on their devices. But this contradicts with the widely adopted stochastic gradient descent (SGD) training procedure, which usually requires data communication for random batch sampling. Federated Learning [12] develops a paradigm for training large models under such situation. Locally trained models are aggregated using FedAvg [19] and then served as initialization model for the next local iteration. The training data confidentiality is achieved by only allowing transferring the model's parameters (rather than the data) under different cryptograph algorithms, which include differential privacy [7], homomorphic encryption [37], block chain [22], etc.

However, recent studies [39, 26, 25] have shown that FedAvg does not provide satisfactory results in the presence of data heterogeneity. A major problem is that all models are designed to fit an "average client" [29], which is difficult when local and global distributions are deviated from each other. This phenomenon is not uniquely observed. Researchers in multi-domain learning have also discovered that directly fitting non-identical domains into a single feature extractor is suboptimal [3]. The leading solution to this problem is to reconfigure the network into domain-agnostic and

35th Conference on Neural Information Processing Systems (NeurIPS 2021).

domain-specific layers [23, 33]. Similar conclusion has also been drawn simultaneously in the field of multi-task learning [27]. The performance is boosted by letting each task choosing task-agnostic and task-specific layers on the fly. Based on such generous observations, we propose a Personalized Federated Learning (PFL) method from the idea of client-agnostic and client-specific initialization. Initialization is considered because it plays the essential role of transferring global knowledge in FL.

Figure 1 dipicts the overall architecture. At the heart of our algorithm is a *mixed initialization* strategy. Instead of fully utilizing the averaged global parameters for initialization, clients will only select a fraction of them, and load the remaining parameters from previous local models. The selection process is decided by a customized loading strategy, which might vary from client to client and time to time. To validate PartialFed thoroughly, we first propose *PartialFed-Fix*, where the loading strategies are inspired by human priors in functionality and classification of different parts of the network. Then we propose an automatic strategy which is learned jointly with the network parameters by gradient descent. We name this algorithm *PartialFed-Adaptive*.

In the case of PartialFed-Fix, our algorithm can be viewed as a federated version of multi-path networks in the multi-domain learning [24, 33]. The shared parameters of different clients are jointly learned by FedAvg, while the client-specific parameters are only learned locally. Treating the network as a combination of global and local blocks makes it possible to learn the knowledge from other clients while keeping the local knowledge stored safely. Although PartialFed-Fix can achieve improved performance, we argue that the fixed loading strategy is a suboptimal solution for PFL since different clients may have different dependencies on the global model. For example, a client might prefer the classifier of the global model at the start of training, which often acquires better generalization ability. In the end, the client is likely to train a local classifier, which brings superior personalized performance. We show that our dynamic algorithm PartialFed-Adaptive is able to capture this change of parameter loading behaviors during the training process.

Despite its simplicity, PartialFed gains surprising performance on many non i.i.d. FL experiments. We construct real world FL experiments by introducing cross-domain classification and detection dataset [30, 33]. For example, on Office-Home dataset, our PartialFed-Fix and PartialFed-Adaptive surpass FedAvg by 4.88% and 5.43% on average accuracy, respectively. Similarly, on UODB dataset, PartialFed-Fix and PartialFed-Adaptive outperform FedAvg by 2.65% and 2.68% on AP50, respectively. More interestingly, the adaptive loading strategy greatly reduce the possibility of clients getting inferior performance caused by the distribution deviation.

## 2 Related Work

**Personalized Federated Learning**  The pioneering work of FedAvg [19] aims to train a global model for all clients. But the goal is hard to achieve when data distributions are non i.i.d.. Wang et al. [31] directly finetune the global model on local dataset for personalized performance. Mansour et al. [18] provide theoretical study of clustering, data interpolation and model interpolation in personalized learning. FedProx [13] introduces a regularization based algorithm. FedCurv [26] and FedCL [35] further consider using Elastic Weight Consolidation (EWC) for parameters importance estimation. T. Dinh et al. [28] theoretically improve the vanilla regularization with Moreau Envelopes. [5, 8] consider using model interpolation to address the PFL problem. Meta learning is also utilized to learn a better initialization for personalized finetuning [6]. The most related works to this paper is parameter decoupling strategies which split networks into global and private layers. Arivazhagan et al. [2] consider global base network with personalized classifiers while FedBN [14] and SiloBN [1] consider private Batch Normalization (BN) layers. The special cases of PartialFed-Fix are equivalent to these methods, and we further examined their combinations and more strategies.

**Multi-Domain Learning**  The problem of training on heterogeneous distributions has been widely studied in multi-domain learning. The difference is that multi-domain learning aggregate data together while FL store data in silos. Our method is related to the recent work [3, 23, 24], which aims to learn a single network across all domains with minimal number of task specific parameters. Bilen and Vedaldi [3] propose to use domain-specific batch and instance normalization while Rebuffi et al. [23, 24] further take series and parallel residual adapters as the domain-specific parts. Wang et al. [33] extend the idea to object detection with SE-ResNet [10] and introduce a multi-domain detection benchmark. Data-driven architectures are also explored in [20, 38, 32] where network pruning or neural architecture search are employed.

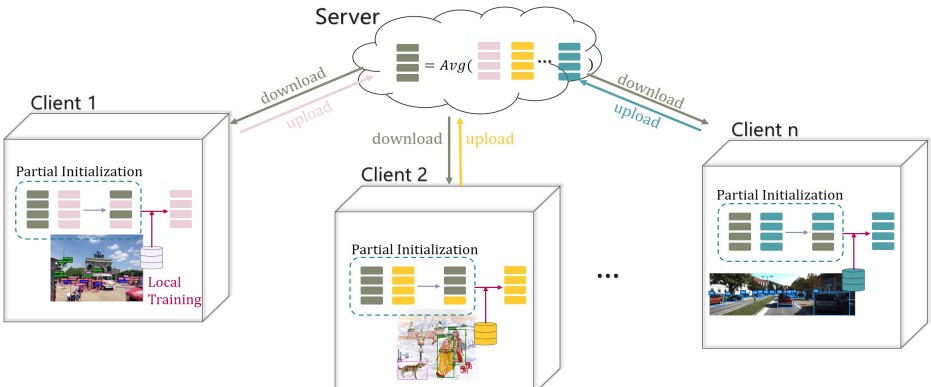

Figure 1: **The snapshot of PartialFed.** In each iteration of federated learning, the central server will send the global model to all clients. Each client will then mix the global parameters from server with local parameters from itself by partial initialization, which is used for the next local training.

## 3 Partially-Loaded Federated Learning

### 3.1 Preliminary

**Federated Learning** We formally describe the problem setup in this section. Assume there are $N \in \mathbb{N}$ clients $c \in C$ who are willing to participate in the joint training. Each client owns a private dataset $\mathcal{D}^c = \{(x_i^c, y_i^c) : i \in \{1, ..., n_c\}\}$, where $n_c$ is the cardinality of the dataset. We use $n = \sum_{c \in C} n_c$ to denote the total amount of data in all clients. The data distribution $P^c(x, y)$ varies from client to client. To be more specific, we do not put any i.i.d. assumptions on either $P^c(x)$ or $P^c(y|x)$, which indicates domain gap and label difference are allowed in the framework.

At the start of each federated iteration $t$, a central server will send a global model $\mathcal{W}_t$ to all clients as parameter initialization. Clients will then use their private data $\mathcal{D}_c$ to train $\mathcal{W}_t$ and get an updated $\mathcal{W}_t^c$. All $\mathcal{W}_t^c, c \in C$ are then sent to server and fused by FedAvg:

$$\mathcal{W}_{t+1} \leftarrow \sum_{c \in C} \frac{n_c}{n} \mathcal{W}_t^c \tag{1}$$

The resulting model is used as initialization for the next federated iteration. When networks are structurally heterogeneous in each client, $\mathcal{W}_t$ denotes the maximum set of shareable parameters. The algorithm will stop on its convergence. Note that we do not incorporate client selection as in McMahan et al. [19], which can be easily added if required.

**PartialFed** Compared to vanilla FL, PartialFed changes the client's parameter initialization process. After receiving global model $\mathcal{W}_t$, each client $c$ will additionally load the local model $\mathcal{W}_{t-1}^c$ from last iteration (See Alg. 1). Instead of purely initializing training with global model $\mathcal{W}_t$, PartialFed uses a mixed initialization, where each parameter is either loaded from global model $\mathcal{W}_t$ or local model $\mathcal{W}_{t-1}^c$. We hope that by partially loading global parameters, client can benefit from sharing knowledge while avoiding local knowledge forgetting caused by FedAvg. The choice of mixing strategy $A_t^c$ will be discussed in section 3.2 and 3.3. All the models are finetuned locally in the last training iteration.

A major advantage of PartialFed is the minimal changes to the original framework. Without sharing gradients, subset of data and customized attributes, clients can enjoy themselves from a highest parameter privacy protocol. In addition, since the server do not need extra operations like knowledge distillation [16] except for FedAvg, the uploaded models are not required to do any forward computation. This enables homomorphic encryption [37] and protects client from information leakage caused by model attacks [36].

---

**Algorithm 1** PartialFed: Partially-Loaded Federated Learning

---

**ClientIteration**$(c, \mathcal{W}_t)$**:**

    Load $\mathcal{W}_{t-1}^c$ from local storage

    Get partial loading strategy $A_t^c$                                                  ▷ partial initialization

    Get partially loaded initialization $\mathcal{W}_{t,init}^c$ by:

$$\mathcal{W}_{t,init}^c[i] = \begin{cases} \mathcal{W}_{t-1}^c[i] & \text{if } A_t^c[i] = 0 \\ \mathcal{W}_t[i] & \text{if } A_t^c[i] = 1 \end{cases}, i \text{ denotes the index of parameter} \qquad (2)$$

    Set $\mathcal{W}_t^c = \mathcal{W}_{t,init}^c$

    **for** each local epoch from 1 to $E$ **do**                                       ▷ regular training

        **for** batch $b \in \mathcal{D}_c$ **do**

            $w \leftarrow w - \eta \nabla \ell(w; b)$, for $w \in \mathcal{W}_t^c$

        **end for**

    **end for**

    Save $\mathcal{W}_t^c$ on local storage

    Return $\mathcal{W}_t^c$

 

**Server:**

    Initialize $\mathcal{W}_0 = [w_{0,0}, w_{1,0}, ..., w_{k,0}]$ at random

    **for** each round $t = 1, 2, \dots$ **do**

        **for** client $c \in C$ **in parallel do**

            $\mathcal{W}_t^c \leftarrow$ ClientIteration$(c, \mathcal{W}_t)$                       ▷ device training

        **end for**

        $\mathcal{W}_{t+1} \leftarrow \sum_{c \in C} \frac{n_c}{n} \mathcal{W}_t^c$                                 ▷ execute FedAvg

    **end for**

---

## 3.2 PartialFed-Fix

We first discuss the fixed manual strategies in this section. Deep networks are often intuitively divided into different functional parts: feature extractor, classifier, etc. These intuitions can be transformed into corresponding loading strategies. An example strategy is shown in Equ.3, where parameters except fully connected layer (*fc*) are loaded from global model. We empirically validate the efficiency by transferring commonly used assumptions to our algorithm in this section.

$$A_t^c[i] = \begin{cases} 1 & \text{if layer } i \text{ is } fc \\ 0 & \text{else} \end{cases} \qquad (3)$$

**"Bottom-up v.s. Top-down"** One of the commonly accepted assumptions in deep network is that blocks close to input image are more related to low-level features while top blocks are associated with high-level concepts. Sharing bottom layers assumes that different clients share similar low-level texture features while the opposite assumes high-level visual concepts are composed analogously. We examine whether PFL training benefits more from sharing global "feature extractor" or global "classifier". To be more specific, we use ResNet18 [9] for these experiments, which has four stages (each with two ResBlock) and a fully-connected layer (Figure 2). A detailed description of the used dataset can be found in section 4.1.

The results are listed in Table 1. Firstly, all experiments with partial parameters loading outperform the full parameter loading, which proves the superiority of the proposed PartialFed. Secondly, there are no significant gap with additional parameters loading from global *s1* to *s1-s4*, but loading global *fc* in this case severely degrades the performance. Thirdly, we find there's consistent accuracy drop from strategy *fc* to *s2-fc*. We hypothesis that sharing too much high-level concepts will hurt the personalized federated learning.

**Batch Normalization** Batch statistics are considered as a key factor when heterogeneous distributions occur and has been widely studied in domain adaptation [15], multi-domain learning [3], and federated learning [1, 14]. We study how the loading strategies of BN will affect the PFL training. Table 2 displays the results on Office-Home dataset. Loading BN locally (i.e. w/o BN) gains improvements on all four domains. Given that the parameters of BN layers are only a very small fraction of ResNet, the improvement is non-trivial. On the other hand, since the BN layer is such brittle in non i.i.d. settings, we hope that a multi-distribution normalization strategy can be proposed in the future.

Table 1: **Bottom-up sharing v.s. Top-down sharing on Office-Home dataset:** The left table shows bottom-up loading strategies, while the right part shows the opposite top-down loading strategies. The column Load describes the partially loaded global layers and *s1-s4* is the abbreviation for loading all global parameters start from stage1 to stage4 (See Figure 2 for ResNet stages). "P", "A", "C", "R" are abbreviations for the domain names. The last row is the result of loading all parameters from the global model (equivalent to *s1-fc*), which is equivalent to vanilla FedAvg with finetuning.

| | | Bottom-up | | | | | | Top-down | | | |
|---|---|---|---|---|---|---|---|---|---|---|---|
| Load | P | A | C | R | mean | Load | P | A | C | R | mean |
| *s1* | 89.46 | 70.70 | 73.33 | 80.53 | 78.50 | *fc* | 88.70 | 71.48 | 72.88 | 81.86 | 78.73 |
| *s1-s2* | 89.46 | 70.12 | 73.44 | 80.64 | 78.41 | *s4-fc* | 87.39 | 71.48 | 73.77 | 80.42 | 78.27 |
| *s1-s3* | 90.54 | 70.70 | 73.33 | 80.53 | 78.50 | *s3-fc* | 87.61 | 70.70 | 71.09 | 80.64 | 77.51 |
| *s1-s4* | 90.22 | 70.51 | 71.21 | 83.52 | 78.86 | *s2-fc* | 88.37 | 68.75 | 70.65 | 82.74 | 77.63 |
| *Full* | 87.99 | 68.70 | 65.85 | 79.22 | 75.44 | *Full* | 87.99 | 68.70 | 65.85 | 79.22 | 75.44 |

Recalling that previous section shows loading *fc* and *s1-s4* simultaneously could harm the performance . We hope this rule can be combined with the BN rule. This leads to the last row of Table 2, where both global BN and global *fc* are not loaded. Surprisingly, this simple modification significantly improves FedAvg from 75.44% to 80.32%, which is 4.88% improvements.

Table 2: Batch Normalization Strategy on Office-Home

| Load | P | A | C | R | mean |
|---|---|---|---|---|---|
| Full | 87.99 | 68.70 | 65.85 | 79.22 | 75.44 |
| w/o *BN* | 88.37 | 72.46 | 71.99 | 82.30 | 78.78 |
| w/o *BN&fc* | 90.43 | 73.44 | 74.33 | 83.08 | 80.32 |

**Skip Loading** Except for the "shared feature extractor - private classifier" network partition, another widely adopted routine in multi-domain and multi-task learning is a loop of "share-private-share-private-..." sub-networks [33, 27]. This paradigm is found to be helpful when tasks interference happens. This paragraph examines the performance of skip sharing with PartialFed.

For simplicity, we label each ResBlock in ResNet18 with letters as in Figure 2. The same number of blocks (4 blocks) is loaded globally for all experiments. The input convolution and *fc* layer are loaded as default. Experimental results are revealed in Table 3. Methods that load adjacent layers (e.g. *AaBb*) get similar mean accuracy except for the *CcDd* strategy. Since *CcDd* is a kind of top-down strategies mentioned above, which does not perform well in general, we infer this is the reason for its inferior performance. Interestingly, the two strategies with skip loading, *ABCD* and *abcd*, get higher performance than all the other strategies, which justify the idea of skip sharing. We attribute the efficiency of skip loading to the fact that clients have the demand of learning client-specific knowledge at all levels of the network instead of only a part of them.

| Load | P | A | C | R | mean |
|---|---|---|---|---|---|
| AaBb | 89.57 | 71.68 | 72.88 | 80.53 | 78.66 |
| AaCc | 89.89 | 72.27 | 71.88 | 80.31 | 78.59 |
| BbCc | 89.78 | 71.09 | 71.76 | 80.75 | 78.35 |
| BbDd | 89.02 | 70.70 | 71.65 | 80.75 | 78.03 |
| CcDd | 87.83 | 69.34 | 69.98 | 81.19 | 77.08 |
| ABCD (Skip) | 90.00 | 73.05 | 72.21 | 81.79 | 79.31 |
| abcd (Skip) | 89.35 | 73.24 | 73.10 | 80.53 | 79.06 |

Table 3: Skip strategy results on Office-Home

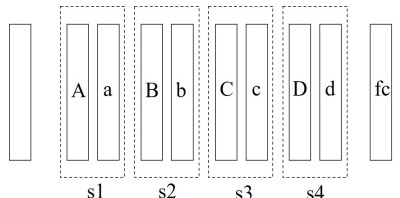

Figure 2: ResNet18 Partition

## 3.3 PartialFed-Adaptive

The hand-designed strategies in PartialFed-Fix are intuitive but not efficient. The search space of all fixed strategies has a complexity of $2^k$ ($k$ denoted the number of layers). If we further consider

the factor of time $t$ and client number $N$, it reaches a complexity of $N \times t \times 2^k$. It's hard to find an optimal strategy by hand in such large space. Therefore, we propose a learnable loading strategy, PartialFed-Adaptive. The main idea is to learn the initialization strategy through a data-driven approach that adaptively chooses which parameters to be loaded globally given the specific client. The objective of the algorithm is to minimize the following loss function on client $c$:

$$\min_{A_t^c} \min_{W_t^c} L_{train}(D_c, W_t^c | W_{t,init}^c = A_t^c(W_t, W_{t-1}^c)) \tag{4}$$

where $A_t^c(\mathcal{W}_t, \mathcal{W}_{t-1}^c))$ is the initialization given by strategy $A_t^c$ and $t$ is the federated iteration. Note that here we only consider the personalized loss for client $c$, but the custom initialization strategy also affects the global training in an implicit way. We leave this problem for future work and focus on the greedy situation in this paper.

The original strategy $A_t^c$ makes discrete decisions, which is not differentiable. In order to optimize strategies and model parameters jointly with gradient descent, we adopt Gumbel-Softmax sampling [17, 11] for modeling strategy $A_t^c$. More precisely, $A_t^c$ is parametrized by a distribution vector $\alpha_t^c \in [0, 1]^{2k}$ with shape $[k, 2]$. Each row $\alpha_t^c[i]$ specifies the probability of loading local and global parameter for the $i^{th}$ parameter $w_i$ and satisfies $\alpha_t^c[i, 0] + \alpha_t^c[i, 1] = 1$. For differentiable training, reparameterization trick is used:

$$u_t^c[i, j] = \frac{\exp\left((\log \alpha_t^c[i, j] + G(j))/\tau_t\right)}{\sum\limits_{k \in \{0,1\}} \exp\left((\log \alpha_t^c[i, k] + G(k))/\tau_t\right)}, j \in 0, 1 \tag{5}$$

where $G = -\log(-\log U)$ is a standard Gumbel distribution with $U$ sampled i.i.d. from $Unif(0, 1)$ and $\tau_t$ is the temperature parameter of softmax at federated iteration $t$. We also use the hard sample trick described in PyTorch document [1] to obtain discrete sampling. The main idea is to substitute soft sample output $y_{soft}$ with hard output $y_{hard} - f_{stop\_gradient}(y_{soft}) + y_{soft}$.

---

**Algorithm 2** PartialFed-Adaptive

---

**ClientIteration-Adaptive**$(c, \mathcal{W}_t)$**:**
    Load $\mathcal{W}_{t-1}^c$ from local storage
    Initialize $\alpha_t^c = \alpha_{t-1}^c$
    **for** each local epoch from 1 to $E$ **do**
        **for** batch $b \in \mathcal{D}^c$ **do**
            Sample strategy $A_b$ by Equ. 5 with parameter $\alpha_t^c$
            Composite batch parameter by:

$$\mathcal{W}_b^c[i] = \begin{cases} \mathcal{W}_{t-1}^c[i] & \text{if } A_b[i] = 0 \\ \mathcal{W}_t[i] & \text{if } A_b[i] = 1 \end{cases}, i \text{ denotes the index of parameter} \tag{6}$$

            Compute loss $l$ with parameter $\mathcal{W}_b^c$
            **if** batch index % $(f_m + f_s) < f_m$ **then**
                $w \leftarrow w - \eta \nabla \ell(w, \alpha_t^c; b)$, for $w \in \mathcal{W}_b^c$            ▷ model parameter update
            **else**
                $\alpha_t^c \leftarrow \alpha_t^c - \eta \nabla \ell(w, \alpha_t^c; b)$                 ▷ strategy parameter update
            **end if**
        **end for**
    **end for**
    Sample best parameter $\mathcal{W}_t^c$ according to $\alpha_t^c$
    Return $\mathcal{W}_t^c$

---

**Training Strategy** The overall learning scheme is summarized in Alg. 2. The model parameters and strategy parameters are iteratively updated by EM algorithm. The updating frequency is denoted as $f_m$ and $f_s$. At every training step, a discrete batch strategy $A_b$ is sampled by Gumbel-Softmax, which guides the composition of global and local parameters. The composited parameter $\mathcal{W}_b^c$ is used to compute loss and either itself or the strategy parameter $\alpha_t^c$ is updated according to training step index. The temperature parameter $\tau_t$ in Equ. 5 is initialized as 5.0 and annealed to 0 as in [27]. The sampling strategy approaches the original discrete distribution with $\tau$ getting close to zero limit [17].

---

[1]https://pytorch.org/docs/master/generated/torch.nn.functional.gumbel_softmax.html

# 4 Experiments

We evaluate the performance of PartialFed, on real-world non i.i.d. tasks including cross-domain classification and detection.

## 4.1 Settings

**Datasets** For federated classification, we use the popular Office-Home [30] dataset, which contains four domains: Art, Clipart, Product and Real World. All domains share the same 65 typical categories in office and home. Each domain has an average of 3k images. Each client use a single domain as its private data, which in total gives 4 clients.

For the detection task, we adopt challenging multi-domain detection dataset UODB [33], which is composed of 11 detection tasks with different domains, categories and data sizes (Table 5). Similar to Office-Home dataset, we take each domain as a client. Except for domain gap, there's also a big difference in the number of training images, from minimum 0.5K to maximum 35K. Note that FedAvg weights each client model by their sample numbers, the severe data imbalance makes it extremely difficult for federated learning.

**Implementation Details** We use PyTorch [21] [2] to implement our algorithms. The detection experiments also use Detectron2 [34][3]. For Office-Home Dataset, we adopt ResNet-18 as backbone network. SE-ResNet-50 [10] is used as the detection backbone, which is the same setting as the original paper [33]. Both models are pretrained on ILSVRC2012 ImageNet [4]. For sampling the best parameters in Alg. 2, we try both hard and soft sampling. Hard sampling takes the parameter that maximize the logits while the soft sampling uses a weighted sum of global and local parameters with the sample logits. Soft sampling is also followed with finetuning. We find soft sampling is better in most cases, so we report the performance of the soft sampling strategy if not specified. For details, we list the hyper parameters in the Appendix.

## 4.2 Cross-Domain Classification

The results on Office-Home dataset are displayed in Table 4. We report the Top-1 accuracy for each domain and on average. On average, the *All* model trained with data from all domains surpasses single domain training, but it does not work well on Product domain. This reflects that naively sharing all parameters is not suitable for all domains even in the non federated training.

**Adaptive Granularity** We experiment adaptive learning at three levels from coarse to fine: stage, block and layer level for ResNet. For example, for the stage-level adaptive, the choice space has complexity of $2^5$, which decides whether each group in $[w_{stage1}, w_{stage2}, w_{stage3}, w_{stage4}, fc]$ is selected globally or locally. The adaptive strategies outperforms fixed ones on average. Among them, the layer level PartialFed-Adaptive works best and outperforms stage-level and block-level by 0.41%. Theoretically, learning at smaller granularity improves the performance upper bound, but also increases the burden for training algorithms. In practice, clients should choose granularity according to its computational and data resources.

**Comparison With State-Of-The-Art** We compare our methods with a set of SOTA algorithms. All of them are reimplemented by referring to the open-sourced code published by the author. Hyper parameters are selected as suggested in the original paper, and the best results are reported. The algorithms are also implemented in a personalized version (i.e. finetune the model at end if needed) for fair comparison. While most existing methods improves the vanilla FedAvg, the best FedBN is still inferior to the single domain training due to domain conflicts. This makes FL unacceptable because the extra communication does not introduce positive impacts. On the other hand, our PartialFed, outperforms both single domain and full data training, which makes it practical in more scenarios.

Although being simple, PartialFed-Fix without global BN and *fc* is already fairly strong on this dataset. It surpasses vanilla FedAvg by 4.88% and FedBN by 1.53% on average accuracy. But it is still affected by domain conflicts and gets only 90.43% on domain Product, which is lower than single domain training. PartialFed-Adaptive overcomes this problem by automatic learning the suitable strategies for each client. It is the only method that beats single domain training on Product domain.

---

[2]licensed under https://github.com/pytorch/pytorch/blob/master/LICENSE
[3]licensed under the Apache License 2.0

This is a thriving result since it improves all clients performance instead of a subset of them. On average, it surpasses vanilla FedAvg by 5.43% and FedBN by 2.09%, respectively.

Table 4: **Office-Home Results**: The first row is the single domain results and the second row is directly training one *All* model with all data. The next block shows vanilla FedAvg and its improvements. For our methods, we first display three representative fixed strategies discussed in section 3.2. Then PartialFed-Adaptive is listed in the last block. Three different granularities are experimented as in the table. We emphasize the best two results of each domain with red and orange.

| Method | P | A | C | R | mean |
|---|---|---|---|---|---|
| Single | 91.59 | 70.22 | 73.77 | 80.42 | 79.00 |
| All | 89.29 | 71.99 | 76.71 | 81.63 | 79.90 |
| FedAvg[19] | 87.99 | 68.70 | 65.85 | 79.22 | 75.44 |
| FedProx[13] | 87.55 | 67.52 | 67.19 | 79.89 | 75.54 |
| pFedMe[28] | 86.24 | 70.28 | 68.19 | 79.56 | 76.07 |
| FedBN[14] | 88.37 | 72.46 | 71.99 | 82.30 | 78.78 |
| PartialFed-Fix | | | | | |
| *s1-s4* | 90.22 | 70.51 | 71.21 | 83.52 | 78.86 |
| w/o *BN&fc* | 90.43 | 73.44 | 74.33 | 83.08 | 80.32 |
| ABCD (skip) | 90.00 | 73.05 | 72.21 | 81.79 | 79.31 |
| PartialFed-Adaptive | | | | | |
| stage | 91.96 | 73.44 | 74.89 | 81.53 | 80.46 |
| block | 91.74 | 73.54 | 74.28 | 82.30 | 80.46 |
| layer | 92.16 | 74.02 | 75.26 | 82.05 | 80.87 |

## 4.3 Cross-Domain Detection

The results of federated detection is summarized in Table 6. The vanilla FedAvg improves performance of clients with relative small datasets: Watercolor and Clipart. But it performs poorly on datasets with special properties, e.g. WiderFace for face detection, DeepLesion for medical image detection. FedBN works well and is able to relief the decline in face and aerial image detection, but it works even worse than FedAvg on DeepLesion, which is distributionally dissimilar to natural images. FedProx does not perform well in this dataset. Restricting contrasting client models to fit the averaged model with regularization term seems harmful given domain gaps at such degree.

For the fixed loading strategy, we first examine SE module and its combination with batch normalization, which is related to the attention mechanism [10]. We also implemented a skip loading strategy similar to the one in section 3.2. We tested two strategies: loading local first block and loading both local first and third block in each SE-ResNet stage. The first one works better on smaller datasets because it provides more knowledge by sharing more parameters. For the adaptive strategy, The block-level adaptive strategy gets best performance in this case. We hypothesis that searching a SE-ResNet-50 strategy at layer-level is too costing and hard to learn, which results in the lower performance of Adaptive-layer.

In general, PartialFed-Fix and PartialFed-Adaptive beats FedAvg by 2.65% and 2.68% and beats FedBN by 0.95% and 0.98%. And again, our PartialFed-Adaptive gets superior performance on distributionally difficult domains, including DeepLesion, WiderFace and DOTA. This proves the effectiveness of our algorithm on extreme domain shift.

Table 5: UODB description

| Datasets | KITTI | WiderFace | PascalVOC | LISA | DOTA | COCO | Watercolor | Clipart | Comic | Kitchen | DeepLesion |
|---|---|---|---|---|---|---|---|---|---|---|---|
| Domain | traffic | face | natural | traffic | aerial | natural | watercolor | clipart | comic | indoor | medical |
| Train | 4k | 13k | 16k | 8k | 14k | 35k | 1k | 0.5k | 1k | 5k | 23k |
| Test | 4K | 6k | 5k | 2k | 10k | 5k | 1k | 0.5k | 1k | 2k | 5k |
| Classes | 3 | 2 | 20 | 4 | 15 | 20 | 6 | 20 | 6 | 11 | 2 |

Table 6: **UODB Results:** The first line is the single domain result. The second line depicts vanilla FedAvg results, which is followed by other competing algorithms. We use *blue* color to highlight domains with large distributional or task deviations, which are challenging for FL algorithms. PartialFed-Fix is displayed with four strategies by considering the network structure. *wo se&BN* denotes for PartialFed-Fix without loading global SE and BN parameters. *fb* and *tb* means the first and third ResBlock in every ResNet stages. Three levels of adaptive methods are also experimented. We emphasize the best two results of each domain with red and orange.

| Method@AP50 | KITTI | *WFace* | VOC | LISA | *DOTA* | COCO | WC | CP | CM | Kit | *DL* | mean |
|---|---|---|---|---|---|---|---|---|---|---|---|---|
| Single | 59.07 | 54.99 | 79.10 | 94.71 | 68.18 | 51.35 | 40.31 | 36.13 | 35.07 | 96.17 | 62.01 | 61.55 |
| FedAvg[19] | 59.91 | 50.99 | 80.54 | 93.39 | 65.04 | 51.74 | 43.19 | 42.81 | 33.46 | 95.41 | 58.26 | 61.34 |
| FedProx[13] | 61.98 | 48.32 | 80.10 | 92.28 | 62.52 | 51.73 | 42.51 | 42.32 | 32.95 | 94.98 | 58.11 | 60.71 |
| FedBN[14] | 62.14 | 52.58 | 81.52 | 94.28 | 67.47 | 51.49 | 49.11 | 43.69 | 39.15 | 94.86 | 57.11 | 63.04 |
| PartialFed-Fix | | | | | | | | | | | | |
| w/o se | 63.16 | 51.34 | 80.97 | 93.44 | 66.10 | 52.45 | 46.89 | 44.86 | 35.96 | 95.36 | 58.43 | 62.63 |
| w/o se&BN | 64.46 | 52.29 | 81.72 | 92.86 | 67.17 | 51.89 | 44.83 | 46.33 | 37.07 | 95.51 | 58.96 | 63.01 |
| w/o fb (skip) | 62.69 | 56.04 | 80.94 | 94.47 | 68.33 | 51.91 | 49.21 | 43.15 | 40.17 | 95.92 | 61.09 | 63.99 |
| w/o fb+tb (skip) | 60.75 | 56.69 | 80.81 | 93.39 | 69.67 | 51.90 | 45.03 | 39.26 | 37.38 | 93.96 | 61.58 | 62.77 |
| PartialFed-Adaptive | | | | | | | | | | | | |
| stage | 61.08 | 55.83 | 81.18 | 93.36 | 68.93 | 51.75 | 48.15 | 44.07 | 38.24 | 95.79 | 63.00 | 63.76 |
| block | 63.44 | 56.04 | 81.40 | 94.25 | 69.07 | 51.89 | 46.74 | 45.41 | 36.50 | 96.34 | 63.15 | 64.02 |
| layer | 62.05 | 54.08 | 80.83 | 94.35 | 68.90 | 51.91 | 45.53 | 44.10 | 38.83 | 95.70 | 62.61 | 63.53 |

## 4.4 Strategy Analysis

We visualize how the strategy changes with FL training, as displaced in Figure 3. Interestingly, there's a clear tendency for all domains to share low-level knowledge than high-level knowledge (i.e. the probability is larger compared to top stages). This coincides with our conclusion in the bottom-up fixed strategies. In the stage1 and stage2, **Clipart** domain differs from other domain, possibly due to its unique image style. For the high-level $fc$ layer, all domains tends to use private parameters at very first of the training except for **Art** domain. Loading local $fc$ layer coincides with our findings in the fixed strategies. On the other hand, we find out that client model with **Art** data learns slower than other domains at the beginning, possibly due to its deviation from the pretrained ImageNet model. Borrowing the knowledge from other domains seems to be a wiser choice in this situation. It's fascinating to see that the adaptive methods is capable of capturing these kind of changes.

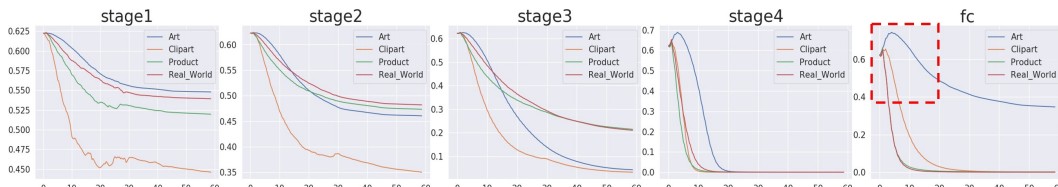

Figure 3: **The change of stage-level strategy across training time.** The $x$ and $y$ axis denotes the federated iteration and the probability of loading global parameters. From left to right exhibits different stages from low-level stage1 to high-level $fc$. Different domains are colored differently. The red dashed box highlight the increase in the tendency of **Art** domain to share $fc$ layer during early training stages.

Our experiments have shown that the adaptive learning strategy has advantages in learning client-specific strategies, which can relief client from negative impacts caused by distribution shift. But it also suffers from the risk of incomplete training, either underfitting or overfitting. For example, we find out that searching a 50 layers ResNet at layer-level is already hard. How to improve the strategy learning efficiency is related to neural architecture search, which considers balance between exploration and exploitation. The data-driven approach also ignores meta-information like datasets cardinality. A possible way to address this is to incorporate human prior into the adaptive algorithm, which is a combination of PartialFed-Fix and PartialFed-Adaptive.

## 5 Conclusion

We demonstrate a simple yet novel approach for personalized federated learning. The core of our method is a mixed initialization, which only partially utilizing the global parameters given by FedAvg. Inspired by commonly accepted priors, we develop a set of manual mix strategies and validate their reliability. To make it further, we develop a data-aware strategy which is adaptively learned with model parameters. Our experiments show that the proposed two strategies, PartialFed-Fix and PartialFed-Adaptive, outperform a set of state-of-the-art methods on cross-domain FL experiments including object classification and detection. More importantly, PartialFed-Adaptive is found to automatically reduce performance degradation caused by extreme distribution heterogeneity.

## Acknowledgments and Disclosure of Funding

We thank Rui Li and Xin Huang for discussions. In the future, we wish to use PartialFed on MindSpore[4], which is a new deep learning computing framework.

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
