# A    Appendix

## A.1    Additional Explanation to PartialFed-Adaptive

This section provides a more detailed explanation for our PratialFed-Adaptive. At the beginning of FL, each client will choose a granularity for its searching. Take stage-level as an example, the network parameters are grouped by stages in ResNet (see Figure 1). Then the client will build its own sampling logits at stage-level to control the loading strategy. A single forward pass is depicted in the figure, which is repeated in every batch of the training. Compared to reinforcement learning algorithms, this implementation has the advantage of differentiability, which ease the training of strategy parameters. But there are still some drawbacks: while the logits of each stage should be considered as equal important, they are in fact weighted by the norm of gradients computed at each feature map. The result is that the training speed of each logit varies from each other. How to give a fair training to all logits is a challenging problem in this framework.

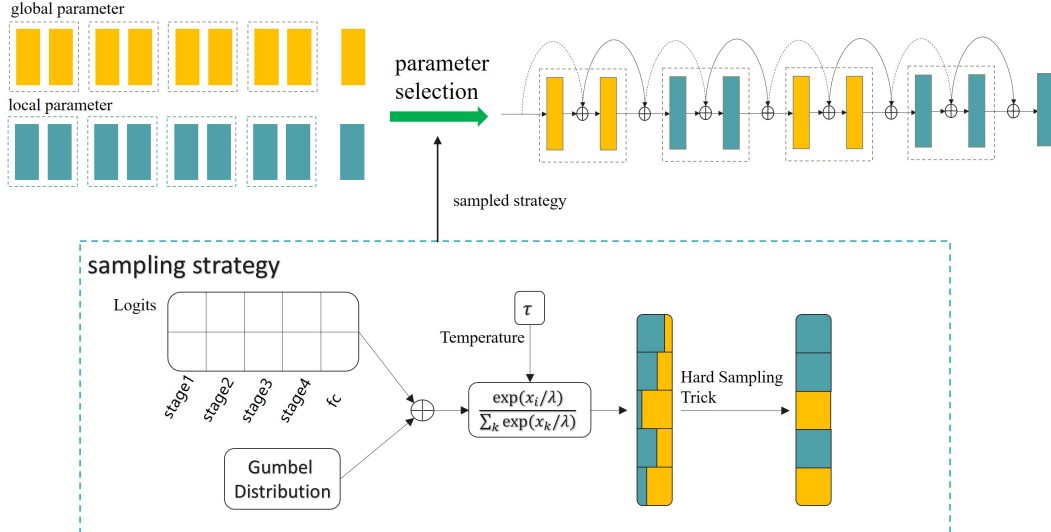

Figure 1: **Computational flow of stage-level PartialFed-Adaptive:** The sampling strategy consists of four steps: 1. sample from a Gumbel Distribution; 2. add the sampled values to unnormalized strategy logits, which controls the probability of loading global and local parameters; 3. a soft strategy is obtained by softmax with temperature; 4. hard sample trick is used to discretize the soft strategy. The sampled strategy is then used to mix global and local parameters. Finally, the combined parameters are used for the ResNet-18 forward pass. The whole process is differentiable, enabling end-to-end training.

# B    Experiments

## B.1    Comparison on Existing Settings

To validate our algorithm more thoroughly, we additionally compare our algorithm in the experimental settings in FedBN [3]. To be more specific, Office-Caltech10 [1] and DomainNet [5] dataset are adopted for comparison. These two datasets contains 4 domains (Amazon, Caltech, DSLR and WebCam) and 6 domains (Clipart, Infograph, Painting, Quickdraw, Real, Sketch), respectively. For Office-Caltech10, all of the ten classes are used for training while for DomainNet, only the top ten common classes are used for experiment. The overall settings are exactly the same as in the open-sourced code published by FedBN. AlexNet with BN layers is used for these experiments. Since there is no stage or block in AlexNet, we only report PartialFed-Adaptive searching at layer-level.

The results are shown in Table 1. Due to the reason that there are relatively less data in each domain, encouraging more parameters to be shared can greatly reduce underfitting of model parameters. FedBN use a very small fraction of domain-specific parameters and share most of them, which makes

it a very strong competitor in this setting. It beats our PartialFed-Fix (wo *BN&fc*) in the DomainNet experiment, which rarely happens in larger datasets (see Appendix B.2 for a comparison with full DomainNet experiments, the w/o *BN* strategy is equivalent to FedBN). But our PartialFed-Adaptive still beats it and gets the best performance.

Table 1: Results on Offce-Caltech10 and DomainNet.

| Method | Caltech-10 | | | | | DomainNet | | | | | | |
|---|---|---|---|---|---|---|---|---|---|---|---|---|
| | A | C | D | W | Mean | C | I | P | Q | R | S | Mean |
| SingleSet | 54.9 ±1.5 | 40.2 ±1.6 | 78.7 ±1.3 | 86.4 ±2.4 | 65.1 | 41.0 ±0.9 | 23.8 ±1.2 | 36.2 ±2.7 | 73.1 ±0.9 | 48.5 ±1.9 | 34.0 ±1.1 | 42.8 |
| FedAvg | 54.1 ±1.1 | 44.8 ±1.0± | 66.9 ±1.5 | 85.1 ±2.9 | 62.7 | 48.8 ±1.9 | 24.9 ±0.7 | 36.5 ±1.1 | 56.1 ±1.6 | 46.3 ±1.4 | 36.6 ±2.5 | 41.5 |
| FedProx | 54.2 ±2.5 | 44.5 ±0.5 | 65.0 ±3.6 | 84.4 ±1.7 | 62.0 | 48.9 ±0.8 | 24.9 ±1.0 | 36.6 ±1.8 | 54.4 ±3.1 | 47.8 ±0.8 | 36.9 ±2.1 | 41.6 |
| FedBN | 63.0 ±1.6 | 45.3 ±1.5 | 83.1 ±2.5 | 90.5 ±2.3 | 70.5 | 51.2 ±1.4 | 26.8 ±0.5 | 41.5 ±1.4 | 71.3 ±0.7 | 54.8 ±0.8 | 42.1 ±1.3 | 48.0 |
| PartialFed-Fix | 58.3 ±1.4 | 44.9 ±1.5 | 88.1 ±1.2 | 91.2 ±3.1 | 70.6 | 48.0 ±1.3 | 25.5 ±0.6 | 40.7 ±1.4 | 71.5 ±0.7 | 56.8 ±0.6 | 38.2 ±0.5 | 46.8 |
| PartialFed-Adaptive | 63.4 ±1.5 | 45.4 ±1.9 | 85.6 ±3.2 | 90.5 ±1.7 | 71.3 | 52.7 ±0.9 | 27.4 ±1.1 | 40.3 ±1.2 | 71.4 ±0.9 | 55.7 ±0.3 | 42.7 ±0.9 | 48.4 |

## B.2 Further Experiments on DomainNet

In section 2, we only include experiments with Office-Home dataset for clear explanation. To further validate the generality of PartialFed, we use DomainNet for further experiments. Different from the settings in Appendix B.1, we use the full training data with 345 classes for our experiments. Table 2 lists all of the results in DomainNet.

**"Bottom-up v.s. Top-down"**    The experimental results are very similar to those in Office-Home. Firstly, all the partially loaded strategies outperforms vanilla FedAvg. Secondly, additionally loading *fc* on the basis of *s1-s4* greatly harms the accuracy. Thirdly, although there is a slight drop, the performance from *s1* to *s1-s4* is close. Given the large gap between domains, the drop seems reasonable. On the other hand, there is a almost linearly decline in the "top-down" strategies (from *fc* to *s2-fc*). We hypothesis that the "top-down" strategies is not a good strategy for cross-domain federate learning.

**Batch Normalization**    Similarly, the w/o *BN* and w/o *BN&fc* improve the FedAvg by a large margin. We are surprising for the consistent improvements of these two simple strategies. But it can be observed that they do not get higher performance than the single domain training in this experiment. With the increase of domain gap and dataset cardinality, more parameters should be made client-specific to fit custom data.

**Skip Loading**    The rule of skip loading also works in DomainNet. We see that *ABCD* still works the best and even outperforms w/o *BN&fc* in this experiment. This further proves the hypothesis that client needs to learn client-specific knowledge at all levels of the network instead of only a part of them. And with the increase of task complexity and domain gap, the skip loading strategy is better than BN strategies, due to its higher degree of freedom. This is also proved in our UODB experiment, where the *fb* strategy works best among all PartialFed-Fix strategies.

**Adaptive**    The adaptive methods also proves its strength in this dataset. The block-level and layer-level PartialFed-Adaptive gets superior performance than the single domain models on mean accuracy. Note that the conflict between domains is severe in DomainNet, where even the *All* model gets inferior performance than the single domain models. It is a tough task for the federated learning to get higher performance than the single domain training.

## B.3 Error Bar

We includes the error bar of the main results in Office-Home in this section. All the experiments are ran by 5 times and compute the mean and standard deviation. See Table 3.

Table 2: **Additional Results on DomainNet:** ResNet18 is used as the backbone network. The first line is the single domain result while the second and third lines represents performance of *All* model trained with all data and vanilla FedAvg. The three kinds of PartialFed-Fix strategies in section 2 is followed by the baselines. The final block displays the results of PartialFed-Adaptive at three granularity. We emphasize the best two results of each domain with red and orange.

| Method | Clipart | Infograph | Painting | Quickdraw | Real | Sketch | Mean |
|---|---|---|---|---|---|---|---|
| Single | 70.96 | 36.74 | 65.40 | 71.07 | 79.36 | 63.91 | 64.57 |
| All | 72.53 | 37.02 | 64.29 | 69.91 | 78.08 | 64.09 | 64.27 |
| FedAvg | 67.23 | 31.90 | 62.32 | 68.06 | 78.65 | 59.31 | 61.24 |
| *PartialFed-Fix "Bottom-Up" v.s. "Top-Down"* | | | | | | | |
| *s1* | 70.62 | 36.16 | 64.19 | 71.20 | 78.79 | 63.59 | 64.09 |
| *s1-s2* | 70.50 | 35.61 | 64.46 | 70.82 | 78.83 | 63.32 | 63.92 |
| *s1-s3* | 70.53 | 35.17 | 64.33 | 70.38 | 78.93 | 62.58 | 63.65 |
| *s1-s4* | 70.07 | 34.02 | 64.36 | 68.99 | 79.25 | 61.51 | 63.03 |
| *Full* | 67.23 | 31.90 | 62.32 | 68.06 | 78.65 | 59.31 | 61.24 |
| *fc* | 71.10 | 36.64 | 64.44 | 71.22 | 78.72 | 64.03 | 64.36 |
| *s4-fc* | 68.74 | 35.02 | 62.61 | 69.71 | 78.27 | 61.23 | 62.60 |
| *s3-fc* | 68.38 | 33.70 | 62.55 | 68.85 | 78.62 | 60.51 | 62.10 |
| *s2-fc* | 67.41 | 32.59 | 62.41 | 68.27 | 78.57 | 60.00 | 61.54 |
| *Full* | 67.23 | 31.90 | 62.32 | 68.06 | 78.65 | 59.31 | 61.24 |
| *PartialFed-Fix with BN strategies* | | | | | | | |
| w/o *BN* | 68.19 | 33.94 | 63.09 | 68.46 | 78.71 | 60.49 | 62.15 |
| w/o *BN&fc* | 70.78 | 35.49 | 64.63 | 69.18 | 79.26 | 62.23 | 63.60 |
| *PartialFed-Fix with Skip strategies* | | | | | | | |
| AaBb | 70.95 | 36.26 | 63.93 | 70.64 | 78.77 | 63.84 | 64.06 |
| AaCc | 71.26 | 36.76 | 64.71 | 70.78 | 78.97 | 63.80 | 64.38 |
| BbCc | 71.22 | 36.07 | 64.60 | 70.49 | 79.01 | 63.45 | 64.14 |
| BbDd | 68.72 | 34.57 | 62.42 | 69.43 | 78.30 | 61.17 | 62.43 |
| CcDd | 68.07 | 33.70 | 62.57 | 68.87 | 78.57 | 60.43 | 62.03 |
| abcd (Skip) | 70.19 | 35.18 | 63.12 | 69.87 | 78.42 | 63.16 | 63.32 |
| ABCD (Skip) | 71.75 | 37.46 | 65.57 | 70.66 | 79.27 | 64.07 | 64.80 |
| *PartialFed-Adaptive* | | | | | | | |
| stage | 71.47 | 36.28 | 64.45 | 70.50 | 79.13 | 63.44 | 64.21 |
| block | 72.22 | 36.90 | 65.19 | 70.61 | 79.42 | 63.89 | 64.70 |
| layer | 72.26 | 37.07 | 65.49 | 70.53 | 79.30 | 64.04 | 64.78 |

# C   Training Details

## C.1   Classification

**Office-Home**   For the Office-Home baseline training, the initial learning rate is set as 0.01 and decay 10 times at epoch 15 and 25. The total epoch is set as 30 epoch. Weight decay and batch size are 5e-4 and 512 for all experiments of this dataset. For all federated experiments, including vanilla FedAvg, FedProx, pFedMe, FedBN and PartialFed-Fix, we train the network for 100 global iteration and each client train their model for 1 epoch in each iteration. We have also tested training the baseline for 100 epoch, but it gets inferior performance due to overfitting. The learning rate decay epoch is 50 and 75 in the 100 global iteration.

For PartialFed-Adaptive, the global iteration is set as 60. Since the size of the dataset is relative small and there are very few steps in each epoch, we set the EM frequency $f_m$ and $f_s$ to 1 full epoch for each client. The local epoch is set as 2. Another epoch of finetuning is also followed after the soft parameter sampling. The initial un-normalized logits for global and local parameters is set as $[0.5, 0]$ to encourage model sharing and the learning rate for this parameter is fixed as 0.3.

Table 3: Office-Home Results with Variance

| Method | P | A | C | R | mean |
|--------|-----|-----|-----|-----|------|
| Single | 91.59 ±0.63 | 70.22 ±0.68 | 73.77 ±0.35 | 80.42 ±0.59 | 79.00 |
| All | 89.29 ±0.41 | 71.99 ±0.69 | 76.71 ±0.64 | 81.63 ±0.26 | 79.90 |
| FedAvg[4] | 87.99 ±0.23 | 68.70 ±0.36 | 65.85 ±0.67 | 79.22 ±0.76 | 75.44 |
| FedProx[2] | 87.55 ±0.28 | 67.52 ±0.42 | 67.19 ±0.67 | 79.89 ±0.27 | 75.54 |
| pFedMe[6] | 86.24 ±0.32 | 70.28 ±0.26 | 68.19 ±0.53 | 79.56 ±0.30 | 76.07 |
| FedBN[3] | 88.37 ±0.22 | 72.46 ±0.29 | 71.99 ±0.50 | 82.30 ±0.35 | 78.78 |
| PartialFed-Fix | | | | | |
| *s1-s4* | 90.22 ±0.31 | 70.51 ±0.22 | 71.21 ±0.50 | 83.52 ±0.66 | 78.86 |
| w/o *BN&fc* | 90.43 ±0.33 | 73.44 ±0.24 | 74.33 ±0.41 | 83.08 ±0.43 | 80.32 |
| ABCD (skip) | 90.00 ±0.21 | 73.05 ±0.38 | 72.21 ±0.37 | 81.79 ±0.44 | 79.31 |
| PartialFed-Adaptive | | | | | |
| stage | 91.96 ±0.40 | 73.44 ±0.19 | 74.89 ±0.39 | 81.53 ±0.51 | 80.46 |
| block | 91.74 ±0.39 | 73.54 ±0.28 | 74.28 ±0.32 | 82.30 ±0.53 | 80.46 |
| layer | 92.16 ±0.34 | 74.02 ±0.32 | 75.26 ±0.62 | 82.05 ±0.67 | 80.87 |

**DomainNet** For DomainNet baseline training, the hyper parameters are the same as in Office-Home. For federated experiments, we set the global iteration at 30 and decays the initial learning rate 0.1 at global iteration 25 with a factor 10. For PartialFed-Adaptive, EM frequency $f_m$ and $f_s$ are both set as 10 steps. 2 local epoch is used to do EM iteration and an additional finetune epoch is followed. All the experiments are trained with one machine with 8 NVIDIA Tesla V100 GPUs. PartialFed-Fix takes an average of 4 hours and PartialFed-Adaptive takes an average of 6 hours for a single experiment.

### C.2 Detection

**UODB** We use Detectron2 to implement our detection algorithms. The domain specific hyper parameters are given in Table 4, Table 5 and Table 6. The initial learning rate is 0.02 and decays two times with a factor or 10. Batch size is 16 for all experiments. For federated learning, the max training steps is doubled for all domain for better convergence. The total federated iteration is set as 30, each domain will communicate at every $\lfloor max\_steps/30 \rfloor$ steps. For the PartialFed-Adaptive, EM frequency $f_m$ and $f_s$ are both set as 10 steps. The learning rate for strategy parameters is fixed as 0.3 for the whole training. In each global iteration, 80% of the local steps are used to do EM update and 20% steps are used to do the finetuning after soft parameter sampling. There are several different settings compared to the original paper: 1. except for the COCO, we use PASCAL VOC 2012 metric to compute AP50 for all datasets; 2. we train on KITTI *train* and test on *val* because KITTI *test* is not included in the published version of UODB. All the experiments are trained with one machine with 8 NVIDIA Tesla V100 GPUs. PartialFed-Fix takes an average of 63 hours and PartialFed-Adaptive takes an average of 102 hours for one experiment.

Table 4: Resize Type: this table gives the used types of resize augmentations. For example, R3 denotes randomly resize the image to 480, 512, 544, 576 or 608.

| Resize Type | Min Size Augmentation |
|---|---|
| R1 | [480, 512, 544, 576, 608, 640, 672, 704, 736, 768, 800] |
| R2 | [640, 672, 704, 736, 768, 800] |
| R3 | [480, 512, 544, 576, 608] |

Table 5: Anchor Type: The left table denotes anchor sizes types while the right table denotes the anchor aspect ratios types.

| Type | Anchor Sizes | Type | Anchor Aspect Ratios |
|---|---|---|---|
| AS1 | [32, 64, 128, 256, 512] | AR1 | [0.5, 1.0, 2.0] |
| AS2 | [6, 8, 12, 16, 24, 32, 48, 64, 96, 128, 192, 240] | AR2 | [1.0] |

Table 6: Domain Specific Hyper-Parameters: This table displays the domain specific hyerparameters. The parameters are selected according to dataset sizes and suggestions given by Detectron2 and UODB benchmark.

| Dataset | Max Steps | Decay Steps | Resize Type | Test Size | Anchor Size | Ratio Type |
|---|---|---|---|---|---|---|
| KITTI | 18000 | 12000,16000 | R1 | 600 | AS2 | AR1 |
| WiderFace | 26000 | 18000,24000 | R1 | 576 | AS2 | AR2 |
| PascalVOC | 18000 | 12000,16000 | R1 | 800 | AS1 | AR1 |
| LISA | 18000 | 12000,16000 | R1 | 800 | AS1 | AR1 |
| DOTA | 26000 | 18000,24000 | R1 | 600 | AS2 | AR1 |
| COCO | 45000 | 30000,40000 | R2 | 800 | AS1 | AR1 |
| Watercolor | 2250 | 1500,2000 | R1 | 600 | AS1 | AR1 |
| Clipart | 2250 | 1500,2000 | R1 | 600 | AS1 | AR1 |
| Comic | 2250 | 1500,2000 | R1 | 600 | AS1 | AR1 |
| Kitchen | 13500 | 9000,12000 | R1 | 800 | AS2 | AR1 |
| DeepLesion | 40000 | 25000,36000 | R3 | 512 | AS2 | AR1 |