# OpenReview forum: "PartialFed: Cross-Domain Personalized Federated Learning via Partial Initialization"
_NeurIPS.cc/2021/Conference — NeurIPS 2021 Poster_

### Official Review · Reviewer_37wQ · 2021-07-04

**Rating:** 6
**Confidence:** 4

**Summary:**

The paper proposes a new method called ‘PartialFed’ to solve the performance degrading caused by data heterogeneity in the federated learning scenario. The main difference between this work and previous work is that the proposed method partially loads global parameters given by FedAvg, while the previous methods load all of the global parameters for each starting point of the local epoch.

For the sampling strategy for the parameters that load the global parameter, two methods are developed in this paper: “PartialFed-Fixed” and “PartialFed-Adaptive”. For the “PartialFed-Fixed”, the parameters are sampled manually based on the commonly accepted assumptions in deep network e.g. low-level knowledge learned from bottom layers. With the "ParitalFed-Adaptive", the sampling strategy is trained data-driven. Gumbel Softmax sampling is used to enable back-propagation through the sampling process.

To validate the proposed methods, cross-domain federated classification and detection are examined. Two partial loading strategies outperform a set of state-of-art methods. Specifically, the “PartialFed-Adaptive” further reduces performance losses due to the extreme distribution heterogeneity and automatically shows the tendency that reflects the assumptions of deep network used for the design of “PartialFed-Fixed”.


**Limitations And Societal Impact:**

It is not clear that the partial averaging idea works for more complicated network architecture.



**Main Review:**

The problem of non-IID data distribution in federated learning is becoming an active field recently. More precisely, this work can be integrated into the personalized layer approach, in which only parameters of some layers can be transmitted to a server and aggregated by FedAvg. Specifically, FedBN, FedPer, and LG-FEDAVG are a form of the “PartialFed-Fixed” sampling strategy. However, this paper provides more ablation studies, so they show that there are better variations on the personalized layering method. More generally, this paper proposes a method that adaptively learns the sampling strategy from data which is clearly different from previous contributions.

That being said, there are several weaknesses.

1) The important task of the work is to find a suitable sampling strategy for personalized parameters, but experiments are carried out with only one network structure for each experiment.
2) In section 3.3, Table 6 shows the experiment result on the cross-domain detection. In this result, the performance of PartialFed-Adaptive is not significantly better than the PartialFed-Fixed method. If we look at each domain respectively, PartialFed-Fixed models are better than ParitalFed-Adaptive models for eight out of 11 domains. Also, for two over three domains with large distributional or task deviations, “w/o fb (skip)” method in PartialFed-Fixed is better than “block-level” PartialFed-Adaptive model. This result can be interpreted in such a way that the proposed adaptive method may not be the optimal choice.
3) I could not see any theoretical or experimental analysis of the convergence of the proposed method, while related work including FedBN offers both.

Some minor comments are as follows.

However, it would be better if the following points were reinforced.
1) There is no description about the metric and its unit written in Table 1,2,3,5 and 6.
2) In line 102 and Equation (3), the notation f_c is used without a description. You can guess it as the fully connected layer before you see Figure 2, but it would be better to define it or to replace it with a word.
3) In line 146 and following lines, the notation k is used without a description.
In addition, following points are typos.
1) In caption of Table 6, “se&BN denotes..” may be “w/o se&BN denotes for..”.
2) In line 172, Equ.4 should be Equ.5.



**Time Spent Reviewing:**

0.5

---

> ### Author Response · Authors · 2021-08-10
> **Authors' Response**
>
> Thank you for your thoughtful review. We hope that our answers can clear out your questions.
>
> **Different architectures in each experiments**
>
> We have included the results of ResNet-18, ResNet-SE-50 and AlexNet with BN layers in different experiments, which to some extent can already prove the generality of our algorithm. But they indeed appears in different experiments. Therefore, we have added a new experiment with ResNeSt-50 [1] with FPN architecture, which is one of the SOTA architectures. The detection parameter setups are amost the same as the official github version except for dataset specifc parameters like number of classes, training steps. We believe that the following results can prove the efficiency of our algorithm on different architectures:
>
> | Method | KITTI | Wface | VOC | LISA | DOTA | COCO | WC | CP | CM | Kit | DL | mean |
> | ---- | ---- | ---- | ---- | ---- | ---- | ---- | ---- | ---- | ---- | ---- | ---- | ---- |
> | Sinigle | 57.30 | 56.72 | 82.44 | 94.81 | 66.98 | 55.95 | 42.70 | 40.16 | 31.42 | 93.76 | 57.6 | 61.84|
> | FedAVG | 61.97 | 56.51 | 83.07 | 94.14 | 68.97 | 57.44 | 57.91 | 58.73 | 44.55 | 93.64 | 58.94 | 66.90|
> | FedBN | 65.47 | 56.52 | 85.80 | 93.93 | 68.21 | 58.89 | 58.71 | 58.50 | 46.87 | 93.50 | 61.13 | 67.96 |
> | Adapive-stage| 64.15 | 58.01 | 84.99 | 95.10 | 69.83 | 58.14 | 60.99 | 58.10 | 44.72 | 93.84 | 62.41 | 68.21|
>
> **Performance about PartialFed-Adaptive**
>
> First, PartialFed-Adaptive performs best on average performance, which is a reasonable metric under the FL setting where multiple clients appear. Second, the significance of PartialFed -Adaptive is that it reduces the reliance on manual selection. For example, while *w/o BN&fc* works best on Office-Home among the fixed strategies, *w/o fb (skip)* works best on UODB dataset. Finding a dataset-specific strategy manually is time-consuming. PartialFed–Adaptive makes it possible to reduce the manual cost from tens of manual strategies to 3 levels of granularity searching. The performance is also superior or at least comparable to the best of PartialFed-Fix. Of course, it is expected that incorporating advanced reinforcement learning and NAS algorithms can increase the overall performance and searching efficiency.
>
> **Analysis of the convergence**
>
> Theoretically discussing the convergence problem of our algorithm is hard, since the network architectures are not restricted and the optimization space is very complex. On the other hand, from the empirical view, we do not observe significant difference in the convergence speed between our algorithm and related works include FedBN. This is somehow reasonable given the similarities of the algorithms. The theoretical discussion will be left as future work.
>
> **Suggestions for clarity**
> 1.	We use top-1 accuracy for all classification tasks and AP50 for detection tasks, the metrics will be made clear on all the tables.
> 2.	Thanks for your reminding, we will add descriptions for the abbreviations.
> 3.	$k$ denotes the dimension of total searchable space, and we will add this to the paper.
> 4.	Thanks for your reminding, we will change the caption of Table 6. And line 172.
>
> [1] Zhang, Hang, et al. "Resnest: Split-attention networks." arXiv preprint arXiv:2004.08955 (2020).

---

> > ### Comment · Reviewer_37wQ · 2021-09-03
> > **Thanks for the clarification**
> >
> > Thanks, authors for the clarification. The experimental results for the different architecture look good, although theoretical results for the convergence are still lacking. Therefore, I will raise my score by 1.

---

### Official Review · Reviewer_iae8 · 2021-07-16

**Rating:** 7
**Confidence:** 5

**Summary:**

This paper proposed an adaptive initialization algorithm for FedAvg, motivated by FedPer in a personalized federated learning scenario. The key contribution lies in the observation of the impact on training accuracy of different initialization strategies and the adaptive initialization algorithm. The empirical result of the algorithm shown significant improvement compared with FedAvg in training with heterogeneous data.

**Limitations And Societal Impact:**

There is no potential negative societal impact of this paper.

**Main Review:**

Originality: Good. The observation of different initialization strategies during local training is impressive, and the way of modeling the impact of the initialization patterns as a part of the loss function was also novel to the FL scenario.

Quality: Good. The empirical results demonstrate the accuracy improvement with different initialization patterns is clear. The method used for updating the adaptive initialization pattern is promising. It has enough comparison between the proposed algorithm with the existing algorithms.

Clarity: Good. The explanation of the dataset and the key observations are easy to understand. The comparisons in the tables are also clear. However, there is some confusion in the algorithm description that needs to be clarified.

Significance: Good. The observation is important in the sense that it provides insight to understanding the impact of each block in NNs and provides a way to do personalized federated learning.

Comment to the paper:

1. In Algorithm 2, step (6) seems to indicate that for E local epochs, the parameter $w$ has not been updated. The algorithm only does $b$ SGD step on $W_b^c$ instead of doing $Eb$ steps. Is it my misunderstanding or the description of the algorithm is incorrect?

2. In other papers, such as FedPD, FedSplit, SCAFFOLD, and FedDyn which optimize local problems to $\epsilon$ accuracy and adaptively regulate the difference between the local and the global model, initialization seems not to be affected by the initialization. Is there any relation between PartialFed-Adaptive and the dynamic regularization methods?


**Time Spent Reviewing:**

2.5

---

> ### Author Response · Authors · 2021-08-10
> **Authors' Response**
>
> Thank you for your thoughtful review, and we are glad that you enjoy the paper. We hope that our answers can clear out your questions.
>
> **Explanation of Algorithm 2**
>
> Yes, $W_b^c$ will not be updated for all the $E*batch steps$. In fact, $W_b^c$ is different in every batch, obtained by a sampled combination of global parameters $W_t$ and local parameters $W_{t-1}^c$. Figure 1 in Appendix A.1 gives an intuitive explanation for the combination process. As a result, the parameter update step in Algorithm 2 will update different combinations of parameters in every batch. For example, for a 4 layers network, if the sampled action $A_b=[0,0,0,1]$ for the first batch, then the first 3 layers of local network and the final layer of global network are selected and updated. In the next batch, $A_\hat{b}$ may change to $[1,1,0,0]$, which leads to a different combination $W_\hat{b}^c$ to be updated. This dynamic combination process is used to explore the best strategy to compose global and local parameters by evaluating a wide range of possible combinations dynamically during the optimization process.
>
> **Relation with dynamic regularization methods**
>
> This is an insightful question and we’ll do our best to discuss it. First, one of the major motivations of adaptive regularization algorithms, including FedPD, FedSplit, SCAFFOLD and FedDyn, is that the local optimization objective $L_i(w) = \sum_k{loss(data_k, w)}$ drifts from the global objective $L_{global} = \sum_{i\in clients}{L_i(w)}$. Under this motivation, novel dynamic regularization algorithms have been developed to encourage the local updates to be in census with global updates. Our PartialFed also assumes that local drifts from global objective. But in addition, we borrow the idea from MDL that the summed version of global objective $L_{global} = \sum_{i\in clients}{L_i(w)}$ is not enough for personalized models, and extra client-specific parameters is required, which transforms the objective to $L_{global} = \sum_{i\in clients}{L_i(w_{global}, w_{local})}$. The PartialFed algorithm aims to find out how to split $w$ into $w_{global}$ and $w_{local}$, while the adaptive regularization algorithms can be used to ensure locally updated $w_{global}$ is aligned with the globally updated $w_{global}$. Eventually, these two kinds of algorithms solves the data heterogeneity probelm from different perspective and are likely to support each other if well designed. We will also add a dicussion for this in the related works.

---

### Official Review · Reviewer_u7s7 · 2021-07-18

**Rating:** 6
**Confidence:** 3

**Summary:**

This paper proposed a novel idea PartialFed, which loads a subset of the global model’s parameters rather than loading the entire model. Specifically, they validate the algorithm with manually decided loading strategies inspired by various expert priors, named PartialFed-Fix. Then they develop PartialFed-Adaptive, which automatically selects a personalized loading strategy for each client.

**Main Review:**

Pros:
-- This paper gives a comprehensive introduction of existing partial loading strategies. The experimental section also provides a comprehensive comparison for these methods with the proposed PartialFed-Adaptive algorithm.
-- The PartialFed-Adaptive algorithm shows certain levels of novelty to learn the initialization strategy through a data-driven approach that adaptively chooses which parameters to be loaded globally given the specific client.

Cons:
 --The PartialFed-Adaptive Algorithm is not clearly explained. e.g., what does $u$ in equation (5) stand for, and how $u$ is used in the optimization process?
-- The algorithm PartialFed-fix is not innovative enough, because there are already some existing works that split networks into global and private layers.
-- According to the experimental results, the layer-level PartialFed-Adaptive works best and outperforms stage-level and block-level PartialFed-Adaptive, but also increases the burden for training algorithms. Thus, layer-level PartialFed-Adaptive must be much more computationally demanding. I suggest that the authors can compare the training time of  PartialFed-Adaptive with different granularity to help achieve a balance between the performance and computational resources.

===
Updates after rebuttal.

The authors provided a clearer explanation of the details of the proposed algorithm in the response, and also added some experimental results to compare the computation resources of algorithms of different loading levels which address my concerns. I raised my score but  I still reserve my opinion that compared with existing works that split networks into global and private layers, the innovation of this paper is still limited.

**Time Spent Reviewing:**

1

---

> ### Author Response · Authors · 2021-08-10
> **Authors' Response**
>
> Thank you for your thoughtful review. We hope that our answers can clear out
> your questions.
>
> **Explanation of PartialFed-Adaptive**
>
> $u_t^c$ in Eq. (5) denotes the sampled action after reparameterization trick, which is used to compose the selected parameters during training. The algorithm has a more intuitive explanation in the appendix A.1, and we’ll refine all the details to ease the understanding.
>
> **Innovation about PartialFed-Fix**
>
> The novelty of PartialFed-Fix is that we re-consider the global-local networks from the more general view of parameter initialization, which then motivates the development of our PartialFed-Adaptive. Additionally, we experiment PartialFed-Fix with different expert priors, which, to best of our knowledge, is not found in existing works. We believe that adding a comparison with existing work in the early sections will make our relative contributions more clear.
>
> **Comparison in training time**
>
> Thanks for the suggestion. For the Office-Home experiments on a machine with 8 NVIDIA Tesla V100 GPUs, the block-level training takes ~4.1 hours, the stage-level training takes ~4.3 hours and layer-level training takes ~4.5 hours. For the UODB experiments, the block-level training takes ~96 hours, the stage-level training takes ~99 hours and layer-level training takes ~105 hours. The training time increases with the dimension of searching space, but the extra costing is acceptable in most cases. We will add the comparison of computation resources in our paper.

---

### Official Review · Reviewer_9V5p · 2021-07-21

**Rating:** 6
**Confidence:** 2

**Summary:**

This paper explores a simple but novel modification to FL approaches to better deal with settings involving non-homogeneous clients. They find that by allowing clients to mix local and global parameters for initializing local training, the added flexibility can notably improve the performance of the respective local end-models. They perform both an exploration of natural fixed mixing strategies and also propose an adaptive mixing strategy that allows for further improvements on average.


**Ethical Concerns:**

None that I have observed.

**Limitations And Societal Impact:**

Yes, I do think the authors have discussed the limitations that come to mind. In terms of societal impact, it does not seem to introduce any new negative outcomes relative to traditional FL.

**Main Review:**

**Originality:** As mentioned in Sec 4, specific instances of separating global/local parameters in FL have been touched upon in previous works. However, this paper certainly seems to usefully generalize such approaches, performing a more thorough empirical investigation into the space of mixing strategies and also introducing a new adaptive strategy.

More on the presentational side though, I do think the manuscript could be more up-front about this relationship with these works (i.e., [2, 1, 14] specifically). To the authors credit, they do mention these citations propose special cases of PartialFed-Fix towards the very end of the paper. However, given this close relationship, I do feel this warrants mention much earlier on, such as in the intro and the relevant subsections of Sec 3.

**Quality:** The results seem solid as a whole. Just had a couple comments here

(1) Given the closeness of the some of the numbers, including information about # of trials and error bars in the main (and also adding them for UODB) would be helpful.

(2) Perhaps not necessary but for completeness, would be interesting to see how the current best PartialFed results compare to SOTA centralized MT/Multi-domain learning results.

**Significance:** The problem seems well-motivated, especially judging by the demonstrated failure of more standard FL methods to be effective on the given datasets (performing worse than fully local learning).  Also, the results seem to indicate that PartialFed is indeed a valuable general strategy, leading to improvements on many of the tasks/clients involved.

**Clarity:** There seems to be some room for improved presentation/clarity. Some things I noticed in particular are:

*  The notation in Eq. (4) seems a bit confusing in that the training loss is minimized over both $A_t^c$ and $\mathcal{W}_t^c$. The latter doesn't really appear in the loss expression?  Also, may be helpful to clarify the $u_t^c$ are the soft samples.

* That final global models are all assumed to be fine-tuned locally could be made more clear, especially in Sec 2. This is mentioned in Sec 3 but in Sec 2, it is only implicitly mentioned in the Table 1 caption.

* Some abbreviations/notation were used without being formally defined such as fc, BN, PFL

* Not central to the paper’s main claims but “Clients can enjoy themselves from a highest privacy protocol” in line 92 does not seem strictly true, given that these approaches may still be liable to downstream inference attacks, i.e. the field has looked at things like differentially private FL to add even more protection.

* Could be nice to move Table 4 closer to the actual results table for UODB for easier cross-referencing


**Time Spent Reviewing:**

6

---

> ### Author Response · Authors · 2021-08-10
> **Authors' Response**
>
> Thank you for your thoughtful review. We hope that our answers can clear out
> your questions.
>
> **An early discussion of relation with existing works**
>
> Thanks for pointing out our problems in the paper arrangement. We will move the related work discussion to the early sections of introduction and Sec 2 as suggested, which will ease the comparison with existing works.
>
> **Error bar in the main**
>
> We do not include error bar in the main due to page limitations. It’s hard to place all the large tables in our main paper. We will try our best to include them in the main pages, or at a later Arxiv version without page limitations.
>
> **Comparison with SOTA MTL/MDL methods**
>
> This is a contributing suggestion given the relation of our work with MTL/MDL. In fact, we are recently trying to re-implement a branch of SOTA MDL/MTL algorithms on datasets including UODB and Office-Home, but this is a time-consuming work, so the comparison will not be appeared in this paper.
>
> **Suggestions for clarity**
> 1.	Eq. (4) should be $\min_{A^c_t} \min_{W_t^c}L_{train}(D_c, W_t^c | W_{t,init}^c=A^c_t(W_t, W_{t-1}^c))$, where $A^c_t(W_t, W_{t-1}^c)$ servers as the initialization of $W_t^c$. We will correct this expression.
> 2.	We will make the training settings more obvious.
> 3.	The abbreviations will be introduced in the later version.
> 4.	Considering the inference attack, maybe “highest parameter privacy” is more accurate in our case.
> 5.	We’ll re-organize the tables to make it clear.

---

### Decision · Program_Chairs · 2021-09-27

**Decision:**

Accept (Poster)

**Comment:**

Thank you for your submission. The reviewers agree that the paper provides an interesting result. The authors should follow the reviewers' suggestions to improve their presentation in the paper.